# Receptor Specificity Engineering of TNF Superfamily Ligands

**DOI:** 10.3390/pharmaceutics14010181

**Published:** 2022-01-13

**Authors:** Fengzhi Suo, Xinyu Zhou, Rita Setroikromo, Wim J. Quax

**Affiliations:** Department of Chemical and Pharmaceutical Biology, Groningen Research Institute of Pharmacy, University of Groningen, Antonius Deusinglaan 1, 9713 AV Groningen, The Netherlands; f.suo@rug.nl (F.S.); xinyu.zhou@rug.nl (X.Z.); R.Setroikromo@rug.nl (R.S.)

**Keywords:** TNF family, protein engineering, receptor specificity, ligand, TNF-α, TRAIL, RANKL

## Abstract

The tumor necrosis factor (TNF) ligand family has nine ligands that show promiscuity in binding multiple receptors. As different receptors transduce into diverse pathways, the study on the functional role of natural ligands is very complex. In this review, we discuss the TNF ligands engineering for receptor specificity and summarize the performance of the ligand variants in vivo and in vitro. Those variants have an increased binding affinity to specific receptors to enhance the cell signal conduction and have reduced side effects due to a lowered binding to untargeted receptors. Refining receptor specificity is a promising research strategy for improving the application of multi-receptor ligands. Further, the settled variants also provide experimental guidance for engineering receptor specificity on other proteins with multiple receptors.

## 1. Introduction

The tumor necrosis factor (TNF) ligand family includes 19 ligands, which each contain a C-terminal TNF homology domain (THD). Each ligand binds to one or multiple TNF receptors (TNFR), containing extracellular cysteine-rich domains (CRD) for ligand binding [1]. The 9 TNF ligands that bind to multiple TNF receptors are TNF-α, LT, FasL, TRAIL, RANKL, APRIL, BAFF, LIGHT, and VEGI [2]. Communication pathways between TNFs and TNFRs are essential for numerous cellular processes, such as apoptosis, immune activation, inflammatory responses, and bone homeostasis, rendering the TNF family of great importance for targeted therapy [3]. Currently, clinical studies on TNF superfamily cytokines are ongoing, such as TNF-α, TNF-related apoptosis-inducing ligand (TRAIL), and receptor activator of NF-kB ligand (RANKL). These studies focus on a diversity of diseases, such as arthritis, malignant tumors, bone metastasis, and osteoporosis [4,5,6,7]. As some of the ligands can bind to different receptors, leading to undesired cross-talk or side-effects, receptor-specific engineering on ligands is crucial for unraveling the mechanism of TNF-related pathways and targeting the ligands to their pharmaceutically relevant receptor. 

This review provides an overview of the engineering of promiscuous TNF ligands towards monoreceptor specificity. From the nine multiple-receptor TNF ligands, six of them are reported to have enhanced receptor specificity by engineering approaches (Figure 1). 

## 2. Tumor Necrosis Factor and Lymphotoxin 

Tumor necrosis factor (TNF or TNF-α) and lymphotoxin (LT) are the first two cytokines that have been characterized as TNF superfamily members. They have a homologous amino acid sequence and were both discovered based on their anti-tumor effects [8]. Moreover, both ligands bind to TNF receptor type 1 and type 2 (TNFR1 and TNFR2) and mediate similar cell signaling transduction. Thus, it is necessary to understand and compare the role of TNF-α and LT in pathways and diseases in order to discriminate the TNFR1- or TNFR2-specific treatments and their medical effects.

### 2.1. Tumor Necrosis Factor and Lymphotoxin

TNF-α was first identified in 1975 based on its ability to induce necrosis in transplanted tumors and the absence of toxicity to embryo in mice [9]. It is mainly expressed by activated macrophages, T lymphocytes, and natural killer (NK) cells [10]. It is synthesized as a cell surface transmembrane protein with 233 amino acids (26 kDa) forming a stable homotrimer, which contains a TNF family homology domain in the extracellular part, a transmembrane domain, a spacer stalk and an intracellular domain [11]. The transmembrane TNF-α (mTNF-α) is subsequently proteolytically cleaved on the spacer stalk by TNF-α-converting enzyme (TACE, or ADAM17) resulting in the release of a 17 kDa soluble TNF-α (sTNF-α) [12]. Both forms are biologically active as homotrimers and capable of binding with TNFR1 and TNFR2, whereas sTNF-α shows a lower affinity to TNFR2 compared with mTNF-α and only activates TNFR1 to mediate cellular signal transduction [13]. 

LT, a cytokine secreted by lymphocytes, was first identified in 1968 based on its in vitro anti-tumor activity [14,15]. Due to its similar effects of LT-α on tumor cells to TNF-α, it was renamed TNF-β in 1985 [16]. However, the discovery of the subunit LT-β revealed its additional functions other than TNF-α and led to the reversion of the name to LT. LT-α is the only TNF ligand without a transmembrane domain and is directly secreted into the cell exterior, while LT-β expresses as a transmembrane protein [17]. Three LT-α subunits form a homotrimer LTα3 that binds to TNFR1, TNFR2, and the herpes virus entry mediator (HVEM). LT-α and LT-β can also assemble the cell surface-bound heterotrimer LTα1β2, which is the predominant form that binds with the lymphotoxin β receptor (LTβR), and the LTα2β1, which binds to TNFR1 and TNFR2 with a minor physiological effect [17,18]. In this section, the LT-related TNFR1 and TNFR2 pathways are discussed, while the HVEM and LTβR pathways are discussed in the LIGHT section. 

It has been demonstrated that TNF-α and LTα3 have similar affinities to TNFR1 and are able to transduce downstream signaling in a similar way [19]. Interestingly, sTNF-α cannot initiate TNFR2, but LTα3, as a soluble form, is able to transduce signaling through TNFR2 [20]. As they have similar amino acid sequences in their TNF homology domain (THD), their different function may be due to differences in their tertiary structure within subunits or the modular assembly mode in trimer formation. 

TNF-α and LTα3 both bind to TNFR1 or TNFR2 to initiate similar signaling transduction; therefore, we will mainly focus on TNF-α-related research with a tacit understanding that LTα3 shows comparable performance. TNFR1 and TNFR2 mediate common and exclusive pathways to induce cell proliferation, inflammation, apoptosis, or necroptosis [21]. Thus, it is important to understand the different roles of TNFR1 and TNFR2 in cell signal transduction.

### 2.2. The TNFR1 and TNFR2 Induced Signal Transduction

TNFR1 protein expression is detected in almost all human nucleated cells, whereas TNFR2 is mainly expressed in the immune system and endothelial tissue [22], such as macrophages [23], T cells [24], monocytes [25], endothelial progenitor cells [26], and mesenchymal stem cells [27]. Both TNFR1 and TNFR2 are transmembrane glycoproteins with four N-terminal cysteine-rich domains (CRDs) in their extracellular domain and a transmembrane domain [28]. Both receptors are capable of activating the nuclear factor kappa B (NF-κB) and activator protein-1 (AP-1) pathway, but TNFR1 also has the capacity to induce cell necroptosis and apoptosis due to the presence of the death domain (DD) [29]. TNF-α/LTα3 binds to TNFR1 to recruit TNFR1-associated death domain protein (TRADD), the receptor-interacting protein 1 (RIP1), the TNF receptor-associated actor 2 (TRAF2) and the cellular inhibitor of apoptosis proteins 1/2 (cIAP1/2). For TNFR2, only TRAF2 and cIAP1/2 are recruited. The complex formation initiates the downstream cascades and ultimately translocates NF-κB and AP-1 to the nucleus to trigger their downstream gene expression, such as FLICE-like inhibitory protein (FLIP). Occasionally, when there is insufficient FLIP, the internalized TNF-α/TNFR1 complex recruits Fas-associated death domain protein (FADD) and procaspase-8 to form the death-inducing signaling complex (DISC) and generate apoptosis signaling [30,31]. Under other circumstances, the deficiency of active caspase-8 results in the binding of RIP3 to RIP1 to assemble necroptosome and induce necroptosis [32].

### 2.3. Engineering TNFR1-Specific Ligands

Due to the different binding affinities between TNF-α and its two receptors, as well as the diverse pathways they trigger, various attempts at specifically targeting the TNFR1 receptor have been made to achieve precise drug delivery, increase effectivity, and reduce TNF-α-related inflammation (Table 1). The research was initially focusing on the construction of mutated TNF-α to target either TNFR1 or TNFR2. In 1993, Ostade et al. constructed two TNF-α muteins, **L29S** and **R32W**, which are specific to TNFR1 [33]. Research showed that even though the artificial R32W had lower affinity to both receptors, it still showed equivalent cytotoxicity to colon and laryngeal cancer cell lines compared with wildtype (WT) TNF-α [33,34]. Then, to regain the full binding capacity of R32W to TNFR1, a double mutant **R32W-S86T** was created, with none binding to TNFR2 [35]. R32W-S86T was widely analyzed and proved to be lethal to tumor cells with low pro-inflammatory effects [34]. Nevertheless, these aforesaid mutants have similar shortages as observed for WT TNF-α, which cause coronary vasoconstriction, hypotension, anti-lipogenesis in vivo, and a short half-life [36,37]. Thus, more efforts were made to solve these problems. By introducing triple site mutations in position 5, 6, and 29 in TNF-α, Atarashi et al. created a mutein, **F4614**, which exhibited higher anti-tumor effects and reduced hypotensive risk more effectively than WT TNF-α [38,39]. Another TNFR1-specific mutein, **M3S,** was made to obtain a higher thermal stability, two-fold prolonged half-life and lower systematic cytotoxicity in vivo. However, the multiple site mutations also resulted in a lower binding affinity to TNFR1 and the M3S did not perform well in animal assays [40]. Recently, a **mutTNF G4** with mutations from sites 84 to 89 was identified, showing higher affinity to TNFR1 than WT TNF-α [41]. The intravenous injection of mutTNF G4 induces the permeabilization of the blood–brain barrier (BBB), which successfully enhances the delivery of the therapeutic reagent into brain tumor.

Investigations of TNF-α antibodies, including both antagonist and agonist antibodies, have been carried out for decades. The antagonist antibody binds to the target protein and blocks its immune responses; conversely, the agonist antibody activates the checkpoint protein and initiates the signaling transduction [43]. FDA-approved antibodies are now commercially available, including infliximab, adalimumab, certolizumab pegol, and golimumab, which are all TNF-α antagonist antibodies [44]. However, the complete blockage of the TNF-α pathway leads to poor results with fatal side effects and low resistance to infections [45]. Therefore, apart from engineering TNF-α variants, the antagonistic antibody specifically targeting TNFR1 attracted more attention. To understand the receptor-specific functions in mice model, **HS1097** and **DMS5540** targeting and inhibiting TNFR1 were made and commercialized [46,47]. In mice model, the application of HS1097 reduces experimental autoimmune encephalomyelitis symptoms, and DMS5540 effectively suppresses collagen-induced arthritis (CIA) without additional effector T cell activation and inflammation reaction [48,49]. Further, a mouse TNFR1 antibody was humanized and named **ATROSAB**. It has a similar affinity to both rhesus and human TNFR1, which makes it a potential agent for the preclinical trials in the CIA model of rhesus monkeys [50]. ATROSAB binds to TNFR1 without signal activation and blocks the activity of both TNF-α and LTα3, leaving the TNFR2 pathway intact [51]. 

In addition to antibodies, small molecules that inhibit specific receptors are gaining popularity due to their low cost and convenient drug administration. Chen et al. utilized pharmacophore model filtering and molecular docking to obtain 10 virtual hits and evaluated their binding affinity to TNF-α and TNFR1 [52]. Three compounds out of ten suppress TNF-α induced cytotoxicity in the noncancerous cell line L929 in a dose-dependent manner. Nevertheless, all the selective compounds inhibit both TNF-α and TNFR1 and thus are lacking receptor specificity. Based on the crystal structure of TNF-α-TNFR1 complex, the TNF-α inhibitor **ZINC09609430**, the TNFR1 inhibitor **ZINC02968981**, and the TNF-α–TNFR1 complex inhibitor **ZINC05462670** were selected [53]. However, the mentioned compounds only underwent simulation prediction; therefore, more biological experiments need to be done to confirm their efficiency. Another strategy was employed to retrieve seven promising compounds targeting TNFR1 by the high-throughput screening of the ChemBridge DIVERSet library. They proved that the selected noncompetitive inhibitors **DS41** and **DSA114** significantly block TNF-α/LTα3-induced NF-κB activation in a TNFR1-specific pattern through perturbing the conformation of TNFR1 without interfering in ligand-receptor assembly [54]. Compared with the development of TNFR1-specific mutants or antibodies, the small molecule inhibitors are still in the early progression stage. 

### 2.4. TNFR2-Specific Applications

TNFR2 has different features to TNFR1 because of its absence of a DD in the intracellular part. The TNFR2-specific mutant protein was mainly used to investigate the receptor-specific signaling pathways and pathogenesis (Table 1), such as, the **D143N-A145R** with extremely low binding to TNFR1 [35]. The mutation sites were selected based on a library of site-directed mutants of human TNF-α, and it was shown that sites 143–145 are responsible for binding with TNFR1 but not with TNFR2. This mutein is frequently used in a comparison with R32W-S86T to explore the functions of different receptors [55,56,57]. Even though these recombinant proteins were made to target TNFR2, more investigations of TNF-α and its receptors have indicated that TNFR2 is incapable of being activated by binding with sTNF-α. 

In contrast to TNFR1, TNFR2 is crucial for the generation and proper functioning of regulatory T cells (Tregs). It regulates immune suppression and promotes apoptosis of autoreactive T cells in multiple diseases [58,59]. Due to the key role of TNFR2 in the immune system, researchers have attempted to develop specific TNFR2 agonists by mimicking mTNF-α. Therefore, based on the achieved TNFR2-specific mutated TNF-α, it was fused with other protein domains to form spontaneous oligomer and to increase the avidity. The fusion protein **STAR2** is constructed by fusing a mutated single-chain mouse TNF-α trimer (D221N-A223R) with one domain of chicken tenascin C (TNC) and three of the fused complexes were trimerized to form a nonameric molecule [60]. STAR2 induced the expansion of Tregs without pro-inflammatory side effects [60]. However, it also showed a mixed result in myocardial infarction mice, which improved the left ventricular function yet decreased the survival rate [47]. In addition, a human TNFR2-specific fusion protein (**TNC-scTNF_R2_**) was created with a single-chain human TNF-α mutant (D143N-A145R) and a human TNC instead of a chicken derivative, which yielded a similar 3D structure to STAR2. It established a neuroprotective effect through preventing oxidative stress-induced cell death from H_2_O_2_ and catecholaminergic in human dopaminergic neuronal cell line LUHMES [61]. Along with using TNC, the dimerization domain EHD2 derived from the heavy chain domain CH2 of IgE was also fused with human TNF-α (D143N-A145R) to form a dimer (**EHD2-scTNF_R2_**) under nonreducing conditions. It was shown that EHD2-scTNF_R2_ protects mice from acute neurodegeneration and memory impairment, and the murine ortholog EHD2-sc-mTNF_R2_ induces the expansion of Tregs with anti-inflammatory responses [58,59]. Unlike TNFR1-specific application, the research into TNFR2 related treatments is still in the early phase. With greater recognition of the crucial role of TNFR2 in neuroprotection and immune system counterpoise, more research is expected to be seen in the future.

## 3. TRAIL

TRAIL, also known as Apo2 ligand (Apo2L), is a homotrimeric transmembrane protein, which has 28% sequence homology with FASL and 23% homology with TNF-α [62]. The membrane-bound TRAIL can be cleaved by proteases into soluble TRAIL that is released into the intercellular space [63]. Moreover, TRAIL selectively induces cell apoptosis in cancer cells and not in normal cells, which causes less side effects as compared to radiotherapy and chemotherapy.

### 3.1. TRAIL Induced Signaling Conduction

There are five receptors known for TRAIL: death receptor 4 (DR4/TRAIL-R1), death receptor 5 (DR5/TRAIL-R2), decoy receptor 1 (DcR1/TRAIL-R3), decoy receptor 2 (DcR2/TRAIL-R4), and the soluble receptor fragment osteoprotegerin (OPG). When TRAIL binds to DR4 or DR5, it recruits FADD to the DR4/DR5 death domain, then DISC is formed to activate caspase 8 through cleaving procaspase 8, to initiate the caspase cascade for cell apoptosis. It can also trigger the intrinsic apoptosis pathway via mitochondrial factors [64]. However, some cells are resistant to TRAIL, which can be attributed to the following: (i) cellular anti-apoptosis proteins, (ii) instability and ubiquitination of caspase protein, (iii) methylation of DR4 and DR5 genes, and (iv) overexpression of decoy receptors DcR1, DcR2, and OPG. Although DcR1 and DcR2 show close homology to DR4 and DR5, they cannot induce apoptosis, because both of them lack an intact or functional death domain. OPG acts as a soluble receptor that is able to sequester extracellularly available TRAIL from DR4 and DR5, thereby also antagonizing the induction of apoptosis.

To evaluate the utility of TRAIL as a cancer therapeutic, human TRAIL (**Dulanermin/AMG 951/RG3639**) was created (Table 2), which induces cell death in glioma, melanoma, Kaposi’s sarcoma, and breast cancer [65,66,67]. It also suppressed tumor growth, which resulted in prolonged survival of CB.17 (SCID) mice bearing breast cancer or colon carcinoma on systemic administration [68,69]. More importantly, it did not cause side effects like neural tissue toxicity in either mice or cynomolgus monkeys and chimpanzees, which could be due to the inability of dulanermin to cross the blood–brain barrier [69,70]. Even though showed promise in preclinical trials, the antitumor effects were not observed in patients. It was speculated that the short distribution half-life and elimination half-life [68,71], the low binding affinity to DR5, and the interference of the decoy receptor are the reasons for the poor performance of the recombinant protein TRAIL in the clinical phase [72]. Engineering longer half-life TRAIL variants that can specifically bind to DR4/5 with lower binding to decoy receptors would solve the problem.

### 3.2. TRAIL Variants with Enhanced Binding to Both DR4 and DR5

**TRAIL G131R** showed enhanced binding affinity to DR4 (2.9-fold K_D_) and DR5 (2.3-fold K_D_) compared to TRAIL WT. Although it also binds to decoy receptors, it still increases the apoptotic activity compared to TRAIL WT in both DR4- and DR5-responsive tumor cells [73]. Another strategy that helped to increase binding to DR4 and DR5 is the membrane-penetrating peptide-alike (TMPPA) technique. Amino acids 114–121 (VRERGPQR) were replaced by RRRRRRRR (named as **TRAIL-Mu3**), allowing faster penetration of pancreatic cancer cells (AsPC-1, BxPC-1, PANC-1) membrane and thereby enhancing apoptosis signaling through stimulating the death receptors distributed on the interior of the cells [74].

### 3.3. TRAIL Variants with Specific Binding to DR4 or DR5

Phage display technology was used to obtain the DR4 or DR5-selective rTRAIL variants **Apo2L.DR4–8** and **Apo2L.DR5–8**. This was the first time that phage display technology was applied successfully for TNF family (Figure 2). In this study, they described Gln-205 and Tyr-216 as important residues for the binding of TRAIL and the five receptors. Compared with TRAIL WT, Apo2L.DR4–8 showed a similar affinity to DR4 and a lower affinity to DR5, while Apo2L.DR5–8 showed the opposite behavior. However, Apo2L.DR5–8 performed better induction of apoptosis in colon carcinoma cell line (Colo205, Colo320) and breast cancer cell line (MDA-MB-231) than TRAIL WT, but Apo2L.DR4–8 did not. This led to the conclusion that DR5 contributes more to cell death signaling than DR4 [75]. Interestingly, it was observed that Apo2L.DR5–8 has lower binding affinity to decoy receptors than Apo2L.DR4–8, which may also be the reason that Apo2L.DR5–8 performs better. Nevertheless, in the same year, another group synthesized **TRAIL.R1-6** and **TRAIL.R2-6,** which are DR4- and DR5-selective rhTRAIL variants. Unlike the previous two variants, the DR4-selective variant TRAIL.R1-6 showed significant apoptosis induction in chronic lymphocytic leukemia and mantle cell lymphoma cells, but not in TRAIL.R2-6 [76]. However, they did not show binding affinity to decoy receptors. These two studies made clear that TRAIL-mediated apoptosis shows preference for DR4 or DR5 signaling depending on the cancer types. 

The second approach is to construct variants that have specific binding to DR4 or DR5 and lower binding to decoy receptors. Our group used the automatic design algorithm FOLD-X approach to successfully obtain DR5-specific TRAIL **D269H/E195R (DHER)**, which has K_D_ = 2.9 nM with DR4 binding, while the TRAIL WT has an 0.17 nM K_D_ towards DR4. DHER maintains high binding to DR5 (0.012 nM) compared to TRAIL WT (0.030 nM). In addition, this mutant shows lowered binding to DcR1, DcR2, and OPG. This variant is also very potent in inducing cell death with a lower median effective dose (ED_50_) in A2789 cells and in Colo205 cells than TRAIL WT [79,81,83]. Further, the combination treatment of subtoxic concentrations of the endoplasmic reticulum (ER) stress-inducing agent 2,5-dimethyl-celecoxib (DMC) and DHER increased TRAIL-induced caspase-8 activation in TRAIL sensitive and insensitive glioblastoma multiforme cell lines (A172 and U87) [84]. Cancer cells treated with artemisinin have higher DR5 expression, which allows better synergy effects of DHER in inducing cell apoptosis [85,86]. DHER also showed promising pro-apoptotic activity in therapy-induced senescent cancer cells or activated hepatic stellate cells [87,88]. For in vivo study, intraperitoneal administration of DHER strongly enhanced DR5 surface expression and caspase-3 activation and delayed A2780 tumor progression with an average reduction of 68.3% [79]. Later, two more DR5 specific variants, **DR5-A** and **DR5-B**, were established with the mixed mutation sites from Apo2L.DR5-8 and D269H. They were demonstrated to be highly selective to DR5 and to not bind to DR4 and DcR1 receptors, also low binding to DcR2 and OPG. Cellular assays showed promising antitumor efficacy independent of the decoy receptor expression level [82]. 

These findings inspired other researchers to combine the preliminary computational screening of proteins to identify positions necessary for TRAIL receptor interactions, which lead to the engineering of DR4 specific TRAIL variant by our group: **D218H** and **D218Y**. These two variants showed a 3.5-fold increase in apparent K_d_ to DR5 than TRAIL WT, while maintaining a relatively consistent K_d_ value to DR4. As for the decoy receptors, D218H and D218Y had a lower binding affinity to DcR2 and OPG, while having a modestly decreased affinity to DcR1. However, both variants caused less cell death in DR4-sensitive cells EM-2 and ML-1 compared with TRAIL WT [77]. Another single mutation **rhTRAIL^DR4^** reversed the TRAIL resistance caused by abundant X-linked inhibitor of apoptosis protein, and accelerated the apoptosis of pancreatic cancer cells [78]. Later, based on the engineering strategy of **rhTRAIL^DR4^**, our group designed a six-site mutated DR4 specific variant **4C7**, which has a higher affinity for DR4 and a lower affinity for DR5. Notably, 4C7 can promote the faster and stronger activation of caspase to induce apoptosis not only in TRAIL-sensitive human colon adenocarcinoma (Colo205, SW948, DLD-1, HCT-15 and CL-34), Burkitt’s lymphoma (BJAB) and the human ovarian carcinoma OVCAR-3, but also the TRAIL-resistant cancer cell line: pancreatic carcinoma (PANC-1) and breast cancer MCF-7 [79]. Further, our group found that HDAC inhibitor can enhance DR4 expression in the DLD-1 cell line, indicating that the combination treatment with 4C7 induces more cell apoptosis [89]. This creates potential for TRAIL as a more general anticancer drug. Later, **rhTRAIL-C3** was created with triple mutations of G131R, N199R, and K201H. Of these mutations, G131R was described as having enhanced binding affinity to both death receptors, and N199Rand K201H were described as leading to lower affinity to DR5. RhTRAIL-C3 was a strong activator of pro-caspase 8 and enhanced the loss of mitochondrial membrane potential, resulting in DR4-mediated apoptosis in TRAIL-insensitive acute myelogenous leukemia and primary blast cell lines at a lower dose (1/6) compared to rhTRAIL WT [80]. This indicates that even though AML cells are completely resistant to rhTRAIL WT, they are sensitive to DR4-specific TRAIL variants, which is similar to the CLL cells described above [76].

Specific binding variants can be used to distinguish which death receptor the primary cells isolated from tumor tissues are more sensitive to. Subsequently, this can be used in the clinic to establish a therapy with a specific variant to target the tumor of the individual patient, which is a step towards personalized medicine. 

## 4. RANKL

### 4.1. RANKL Induced Signaling Conduction

RANKL is a soluble protein secreted by bone-producing osteoblasts and can bind to its receptor RANK, which is expressed on bone resorbing osteoclast precursor cells. RANKL can also bind to the soluble OPG and same as in TRAIL, OPG will compete for the binding between RANKL and RANK [90]. The RANKL–RANK–OPG system was first found to play a major role in bone remodeling systems to control the bone producing and resorbing process, where the binding of RANKL recruits TNF receptor-associated factors 1, -2, -3, -5 and -6 to intracellular domain of RANK, leading to the expression of osteoclast-specific genes, like NF-kB, tartrate-resistant acid phosphate (TRAP), cathepsin K, and matrix metallopeptidase 9 (MMP-9). This facilitates the fusion of the osteoclast precursors into osteoclast, resulting in bone resorption [91,92,93]. If the RANKL and RANK pathways are overactivated, excessive osteoclasts leads to osteoporosis. In addition, MMP-9 is a significant enzyme to degrade extracellular matrix (ECM), which is the main composition of fibrotic tissue [94,95]. Interestingly, the binding of RANKL and RANK contributes to the degradation of ECM and prevents fibrosis [96]. 

### 4.2. RANKL Variants with Lower Binding to RANK

Protein engineering has been used to develop RANKL variants with lower binding affinity to RANK to prevent osteoporosis (Table 3). Before the crystal structure of RANKL–RANK was confirmed, mutation engineering of RANKL was conducted based on the crystal structures of TNF-β–TNFR and TRAIL–DR5 [97,98]. The solvent-accessible surface loops of RANKL are unique, and are distinguished as the AA″ loop (residues 170–193), the CD loop (residues 224–233), the DE loop (residues 245–251), and the EF loop (residues 261–269). The AA″ loop and DE loop were thought to be RANKL–RANK binding related. From these loops, **I248**, which is located in the DE loop of RANKL, was selected. **I248D** showed an 8-fold decrease of osteoclast-inducing activity to RANKL WT, but no protein–protein interaction data was shown [99]. After identifying the 3D crystal structure of the RANKL–RANK complex, more precise predictions can be made for the mutation development [90]. As the hydrophilic interactions dominate the binding between the DE loop of RANKL and RANK, our group substituted other amino acids on the I248 site according to the change of binding energy and hydrophilic properties. We built up a computer-aided structure-based RANKL mutant library, in which **I248K** and **I248Y** showed higher binding association rate constant (K_a_) to RANK with the respective K_a_ of 15.7 and 17.2 M^−1^s^−1^, while RANKL WT was 4.3 M^−1^s^−1^. Meanwhile, the dissociation rate constant (K_d_) is similar to that of RANKL WT, which results in the mutants having four times the affinity of the wild type. However, higher affinity did not increase RANKL-induced osteoclastogenesis in RAW 264.7 cells; rather, they reduced it [100]. The general workflow of the computer-aided protein engineering is depicted in Figure 3. Some studies demonstrated that binding of the ligand to receptors limits the conformational freedom of the receptor and thereby hinders the intracellular signaling [101,102]. This may also apply for the case of RANKL–RANK. A follow-up study of the AA″ loop after the resolving of RANKL–RANK crystal structure was also completed. **rRANKL5** is a recombinant RANKL with deletion sites from aa 246 to 318 in the AA″ loop and CD loop. SPR data showed that the mutant rRANKL5 bind less to RANK compared to RANKL WT and inhibit the osteoclastogenesis progress in RAW 264.7 cells. However, they could not determine the K_D_ [103]. 

### 4.3. RANKL Variant with Lower Binding to OPG and Targeted to RANK

To maximize the binding of RANKL to RANK, mutants were created that are not able to bind to OPG. As mentioned above, binding of RANKL and RANK prevent fibrosis by the degradation of ECM. High level of OPG is found in several fibrotic tissues in the liver, vascular system and lung, which correlate with profibrotic effects [105,106,107,108]. From our RANKL mutant library, **RANKL Q236D** showed reduced binding to OPG compared to RANKL WT. The binding of Q236D to OPG showed a 3-fold decrease of K_a_, and a 10-fold increase of K_d_, which led to a 30-fold lowered binding affinity K_D_ to OPG. Binding simulation performed by BIOVIA Discovery Studio 4.5 showed that changing the RANKL residue GLN to ASP at 236 position eliminates the intramolecular hydrogen bonds between residue E93 and E95 of OPG (Figure 4). Meanwhile, the binding of Q236D to RANK is as strong as RANKL WT to RANK. This mutant Q236D is still able to activate the formation of osteoclasts. Further, the mutant can significantly increase the gene expression of MMP9, which is important for ECM degradation. Therefore, the research of RANKL Q236D provides us a new therapeutics strategy against fibrosis [104].

### 4.4. RANKL-Related Peptides and Antibodies That Inhibit RANK Funtion

In addition, there are some studies focusing on peptides that act as an inhibitor of RANK through mimicking RANKL. Among them, **MHP1**, a mimic of DE loop and part of EF loop of RANKL, does not activate the NF-kB signal, blocked RANKL-induced osteoclast differentiation in a dose-dependent manner [109]. Another peptide (**WP9QY**) blocked bone resorption by inhibiting recruitment and activation of osteoclasts to prevent osteoclastogenesis in mice [110]. **Peptide L3-3**, which is based on crystal structure of extracellular domains of mouse RANK–RANKL complex, strongly bound to RANKL ectodomain to block the RANKL-induced maturation of osteoclast precursors [111]. The monoclonal antibody denosumab also inhibits RANK–RANKL interactions and limits the formation and function of osteoclasts, leading to the suppression of bone resorption [91,112]. Denosumab is currently in multiple clinical trial phases for the treatment of bone-related diseases and cancers, such as osteoporosis, breast cancer, and prostate cancer [113,114,115,116].

## 5. FASL

FasL (CD95L/Apo1L) is a TNF-related homotrimeric transmembrane protein, expressed on diverse immune cells like B, T, and NK cells [117]. It conducts cell apoptosis signaling after binding to a membrane-bound Fas receptor. The receptor has three CRDs in the extracellular part, which are required to bind to FasL. The intercellular death domain is crucial for conduction of apoptosis signaling [118,119]. FasL also interacts with decoy receptor 3 (DcR3). DcR3 is a soluble receptor secreted in the extracellular space, which can compete with Fas for the binding to FasL and thereby is able to suppress the FasL–Fas mediated apoptosis [120].

Binding of membrane-bound FasL to Fas recruits the adapter molecule FADD to the death domain; this, in turn, can activate procaspase 8 in a similar way to TRAIL signaling [121]. Fas is widely expressed in not only memory and effector T cells upon contact with antigen, but also in nonlymphoid tissues, such as the skin, liver, ovary, lung, and heart, and various tumor cell lines [122,123,124]. As FasL is also mainly expressed on T cells, activation of cell death signaling eliminates the redundant T cells after an immune response, pathogen infection, or in oncogenically transformed cells [125]. However, the metalloprotease-cleaved soluble form of FasL (sFasL) cannot form the DISC complex and therefore is not able to induce apoptosis signaling [126].

sFasL fails to activate Fas-induced signaling, but oligomerization of ligand could reverse this functional defect. Several fusion proteins of sFasL were engineered to overcome this problem by self-oligomerization. One such self-oligomerizing FasL fusion protein is **APO010** (**MegaFasL**), which is a hexametric fusion protein created by the fusion of two trimeric FasL extracellular domains to a dimer-forming stalk region of human adipocyte complement-related protein (which itself has no functional activity) [127,128]. APO010 showed anticancer activity in mice carrying various types of cancer cells, such as multiple myeloma, colon, and glioma cancers without obvious side effects [129,130,131]. APO010 is currently in Phase I clinical trial in patients with solid tumor; patient recruitment is complete, but new data are not yet available [132]. Another group fusion protein is engineered with tumor-associated antigen. The first fusion protein is **CTLA-4–FasL**, which was designed to fuse cytotoxic T lymphocyte-associated antigen-4 (CTLA-4) and FasL amino acids 127–281. CTLA-4 binds to membrane-bound ligand B7 on antigen-presenting cells (APC), which helps FasL act like mFasL to trigger apoptosis. CTLA-4–FasL, as a representation of “trans signal converter proteins” strategy, efficiently inhibits Jurkat cell growth [133]. Another protein–protein fusing form is **CD40·FasL**, in which CD40 functions as apoptotic receptor [134]. Individual CD40·FasL or the combination with CTLA-4–FasL induce apoptosis in malignant cells [135,136]. Further, FasL fused with a single-chain variable fragments (scFv) worked well. scFv is an engineered antibody targeting specific antigens that are expressed in transformed cells. Once scFv binds to specific cells, the scFv-fused FasL performs as the membrane-bound FasL and induces cell signaling transduction [137]. **sc40-FasL** was made by fusing with sc40, which targets fibroblast activation protein (FAP). This compound achieved the local activation of FasL to suppress FAP-positive tumor cells. This is the first described cell-surface antigen-mediated local activation of Fas in vivo [138,139]. Another three scFv fused protein, **scFvRit:sFasL**, **scFvCD7:sFasL** and **cc49scFv-FasL_ext,_** also showed promising anttumor activity [140,141,142].

However, all research on the recombinant FasL has focused on mimicking the function of membrane-bound FasL to interact with receptor Fas. DcR3 has been found to be overexpressed in inflammatory diseases and cancer. Excessive DcR3 can occupy FasL to limit the FasL-induced signaling conduction. Recombinant oligomerized FasL is expected to overcome DcR3 blockage. Therefore, based on the experience of other TNF-family ligands with multireceptors, such as STAR2 and TNC-scTNFR2, we expect more studies to enhance the selection specificity towards Fas or DcR3.

## 6. LIGHT

### 6.1. LIGHT and Its Receptors

LIGHT (TNFSF14) was first identified by its binding ability to HVEM:Fc in HEK293 cells in 1998; it shows a high sequence homology with LT-β and the other TNF family ligands [143]. It is also expressed as transmembrane protein, which only forms homotrimer from 29 kDa monomers. Like TNF-α, it exists in both soluble and membrane bond forms and mainly expresses in immune cells, such as activated T cells, natural killer (NK) cells, neutrophils, and dendritic cells (DC) [144]. As well as HVEM, LIGHT also binds to LTβR and decoy receptor 3 (DcR3). These three receptors all have four CRDs in their C-terminal domain that interact with LIGHT [145,146,147].

Both HVEM and LTβR are transmembrane protein belonging to the TNFR superfamily. HVEM mainly expresses in lymphocytes, NK cells, epithelium cells, and monocytes. In contrast, LTβR is not expressed on lymphocytes, but rather on follicular dendritic cells (FDC), myeloid cells, monocytes, and DC [148]. By interacting with LIGHT or LTα3, HVEM recruits TRAF2 and TRAF5 and further induces NF-κB and AP-1 pathways [149,150]. The interaction of ligands with HVEM enhances antibacterial activity of monocytes [151], blocks glycoprotein D of herpes simplex virus (HSV) entry [152], and activates NK cells and CD8^+^ cells [153]. As a receptor for LIGHT and LTα1β2, the formation of a ligand-LTβR complex results in the shuttling of TRAF2, TRAF3, and TRAF5 to the locality of LTβR and mediates noncanonical NF-κB signaling, AP-1 signaling, or cell apoptosis (even without DD in its intracellular domain) [154]. Hence, LTβR signaling induces tumor cell death in a reactive oxygen species (ROS) dependent way, limits the capacity of tumor cells recruiting immune suppressor cells, assists the extravasation of leukocyte to acting locus, regulates immune cell migration, and maintains splenic microenvironments [148,155]. DcR3 is different from the other two receptors due to its lack of transmembrane domain, making it an inherent inhibitor of LIGHT [156]. Due to its divergent functions from HVEM and LTβR as well as the existence of DcR3, the receptor-specific targeting strategy is of vital importance. 

### 6.2. Specific Targeting HVEM and LTβR Strategy

Similar to previously employed strategies, to remove the neutralizing influence of DcR3, mutated ligands were developed and their biological activity was determined. Morishige et al. constructed a phage library to select a receptor-specific LIGHT mutant with a low affinity to DcR3. Five muteins were selected, of which **Clone 1** with four site mutations shows the strongest binding affinity to both HVEM and LTβR with high equilibrium dissociation constant to DcR3. Its antitumor capacity is higher than WT LIGHT in the presence of DcR3 in HT29 [157]. They also demonstrated that the G–H loop motif (222 to 229) is crucial for the interaction between LIGHT to HVEM, LTβR, but not to DcR3, providing a clear understanding of the association between them.

With the application of a receptor-specific biologic agent, the diametrically opposed functions of HVEM and LTβR have been discovered. LIGHT G119E mutant has a high affinity to HVEM, but fails to bind LTβR, cannot induce HT29 cell death [158]. In addition, LTβR agonist **31G4D8** (antibody) and LIGHT-R228E mutein induces ROS production, resulting in cancer cell apoptosis [155]. Recently, it was shown that LTβR agonistic antibody **BS1** initiates apolipoprotein B mRNA editing enzyme catalytic subunit 3B (APOBEC3B) expression through noncanonical NFκB signaling to suppress hepatitis B virus (HBV) replication [159,160,161]. 

## 7. Summary

In this review, we described the engineering of six promiscuous TNF family ligands towards single receptor specificity. Studies of the remaining three ligands, BAFF, APRIL, and VEGI, are still in the early stage due to difficulties in protein expression and purification. 

Currently, antibodies are also used for targeting a specific receptor. Compared with antibodies, the engineered TNF ligand variants involve fewer residue changes, which leads to less complicated interaction kinetics, reduced autoimmune response, and higher treatment safety in vivo. Further, as variants can be produced by *E. coli*, they are easier to produce and cost effective. However, the use of ligand variants can be hindered by drawbacks, such as expression and purification difficulties, low stability, and short half-life. Our group is making several attempts to solve these problems, such as the long-term secretion of adenovirally expressed systems.

In summary, the specificity of receptor selection is helpful to (i) investigate specific signaling pathways, such as D143N-A145R was used to study the TNFR2-specific signaling pathway, (ii) reduce side effects, such as TNFR1-specific ligand R32W-S86T, which was found to be lethal to tumor cells with low proinflammatory effects, and (iii) increase efficacy of ligands, such asDR5-specific ligand DHER, which induced more cancer cell death than TRAIL WT by escaping from the neutralization of the decoy receptor OPG. Therefore, improving receptor specificity is a promising research strategy for multireceptor ligands.

## Figures and Tables

**Figure 1 pharmaceutics-14-00181-f001:**
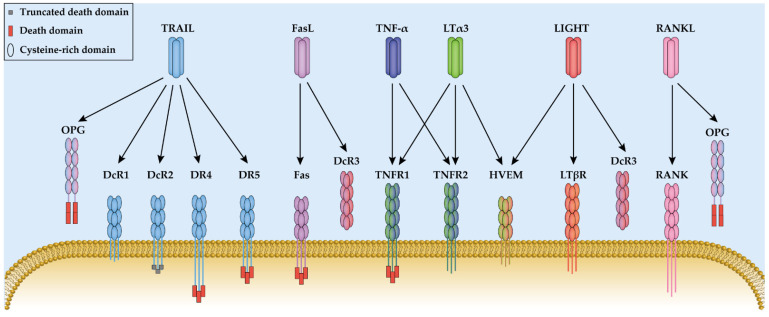
Interactions between multi-receptor TNF superfamily ligands (top) and their receptors (bottom). The ligands are composed of three monomers with a jelly-roll fold. We show those ligands that have been subjected to receptor-specificity engineering approaches. TNF homology domains are shown as a cylinder, the cysteine-rich domains as oval shape, and the death domains as red box. DcR2 has a truncated death domain shown as gray box.

**Figure 2 pharmaceutics-14-00181-f002:**
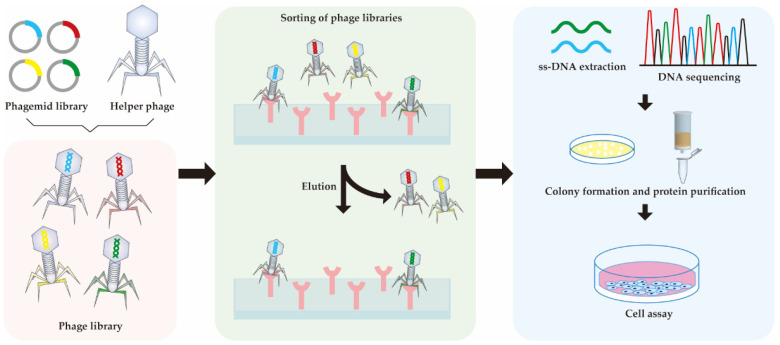
Workflow of phage display, which is used for the high-throughput screening of protein interactions. DNA encoding protein of interest are cloned into the genomes of multiple helper phages. The protein of interest, which is displayed on the phage surface, binds to the immobilized receptor. After the binding and elution steps, phages fractions with high binding affinities are collected and the DNA is sequenced. Amplification and purification are followed for further analysis.

**Figure 3 pharmaceutics-14-00181-f003:**
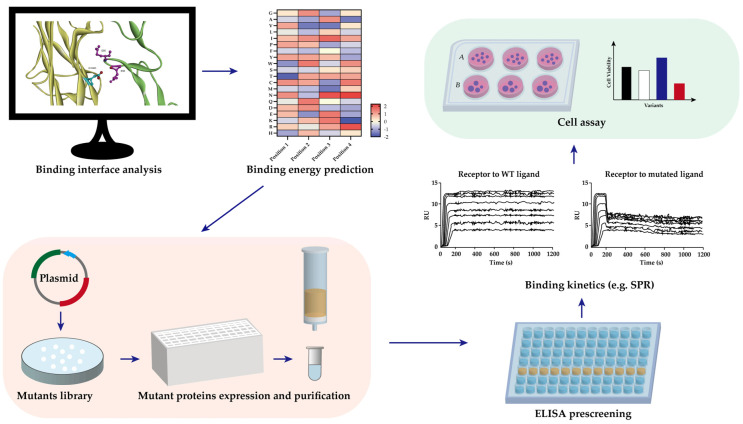
General workflow of computer-aided protein engineering, which helps the design and optimization of the target protein.

**Figure 4 pharmaceutics-14-00181-f004:**
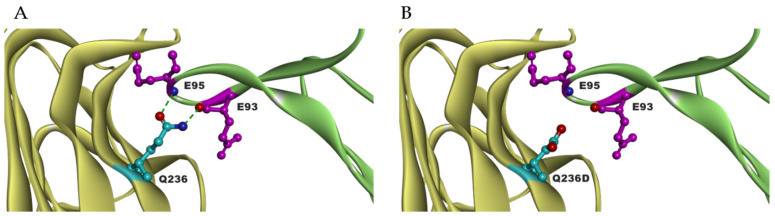
Predicted area of interaction of RANKL and OPG receptor (PDB:4E4D) around position 236. (**A**) OPG (green) binds to RANKL WT (yellow); (**B**) OPG (green) binds to RANKL Q236D (yellow).

**Table 1 pharmaceutics-14-00181-t001:** The mutation sites and binding affinity of the receptor-specific TNF-α variants.

Ligand	Specificity	Variants	Mutation Sites	Binding Affinity (nM)		Ref.
				TNFR1	TNFR2	
TNF-α	TNFR1 &TNFR2	WT	/	15.8	35.3	[41]
	TNFR1	L29S	L29S	− ^1^	−	[42]
		R32W	R32W	−	−	[42]
		R32W-S86T	R32W/S86T	3540	NB ^2^	[42]
		F4614	T5G/P6D/R29V	−	−	[38]
		M3S	L29S/S52I/Y56F and 449/455del	−	−	[40]
		mutTNF G4	A84S/V85S/S86T/ Q88N/T89P	8.72	NB	[41]
	TNFR2	D143N-A145R	D143N/A145R	NB ^2^	13.1	[42]

^1^–means not available. ^2^ NB means no binding.

**Table 2 pharmaceutics-14-00181-t002:** The mutation sites and binding affinity of the receptor-specific TRAIL variants.

Ligand	Specificity	Variants	Mutation Sites	Binding Affinity (nM)		Ref.
				DR4	DR5	
TRAIL	DR4 & DR5	G131R	G131R	8.7 ± 1.0	7.9 ± 1.3	[73]
		TRAIL-Mu3	aa 114–121 (VRERGPQR) were replaced by RRRRRRRR	− ^1^	−	[74]
	DR4	Apo2L.DR4–8	Y213W/S215D/Y189A/ Q193S/N199V/K201R	2.3-fold to WT	NB ^2^	[75]
		TRAIL.R1-6	Y189A/Q193S/N199V/ K201R/Y213 W/S215N	−	−	[76]
		D218H	D218H	12.3 ± 0.6	28.6 ± 1.7	[77]
		D218Y	D218Y	107 ± 0.4	23.3 ± 0.4	[77]
		rhTRAILDR4	S159R	0.37 ± 0.12	4.3 ± 0.9	[78]
		4C7	G131R/R149I/S159R/ N199R/K201H/S215D	0.021 ± 0.01	7.21 ± 4.2	[79]
		rhTRAIL-C3	G131R/N199R/K201H	−	−	[80]
	DR5	Apo2L.DR5–8	Y189Q/R191K/Q193R/ H264R/I266L/D267Q	NB	0.8-fold to WT	[75]
		TRAIL.R2-6	Y189Q/R191K/Q193R/ H264R/I266L/D267Q	−	−	[76]
		D269H/E195R	D269H/E195R	2.9 ± 1.7	0.012 ± 0.005	[81]
		DR5-A	Y189N/R191K/Q193R/ H264R/I266L/D267Q/D269H	NB ^2^	0.33 ± 0.005	[82]
		DR5-B	Y189N/R191K/Q193R/ H264R/I266L/D269H	NB	0.71 ± 0.013	[82]

^1^–means not available. ^2^ NB means no binding.

**Table 3 pharmaceutics-14-00181-t003:** The mutation sites and binding affinity of the receptor-specific RANKL variants.

Ligand	Specificity	Variants	Mutation Sites	Binding Affinity (pM)		Ref.
				RANK	OPG	
RANKL	RANK	I248D	I248D	− ^1^	−	[99]
		I248K	I248K	9 ± 2	−	[100]
		I248Y	I248Y	8 ± 3	−	[100]
		rRANKL5	aa 246–318 deletion	−	−	[103]
		Q236D	Q236D	15.0 ± 3.2	112.3 ± 24.4	[104]

^1^–means not available.

## Data Availability

Data sharing not applicable.

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
