# Peer review of "Receptor Specificity Engineering of TNF Superfamily Ligands"

_pharmaceutics, 2022, doi:10.3390/pharmaceutics14010181_

Round 1

Reviewer 1 Report

The authors have written a review describing the TNF family ligands and their receptors, particularly elaborating on ligand engineering for receptor specificity. The text is well written and explanations are comprehensive. My main criticism is to elaborate the introduction to broadly explain about the TNF superfamily before going into details and also include cartoons to help readers follow the text better.  

  1. The introduction does not introduce the TNF family. The authors have started by directly explaining about TNF ligands and receptors. First, the authors have to write about TNF super family which are cytokine receptors that bind ligands through an extracellular cysteine rich domain. Explain, significance of this superfamily and why studying it is important. Also explain expression pattern (how these receptors are expressed in many tissues) and how that is relevant for clinical research. Then, go into details on ligand and receptor. 
  2. The authors have to include a cartoon figure of all TNF ligands and receptors. Also include broad downstream pathways of each receptor. 
  3. A cartoon of domains in receptor and ligands will also be useful. Particularly useful to know the similarities and differences in structure between them especially in the extracellular ligand binding and intracellular signaling regions. 
  4. Cartoon to show different mutations and how they affect structure and function of receptors will also be useful to readers. 

Reviewer 2 Report

Dear Authors,

presented manuscript is very interesting, but I thing, that, apart from organizing the topic of TNF-family, it doesn't bring anything new. In my opinion the Authors did not highlight, how Their analysis differs from other so far published on this topic.

Moreover in the "Discussion" section, the Authors mainly summarize the information presented in the article, without any “disscussion”.

Manuscript is particularly illegible in context of the presented references- in fact  it's not possible to confront the presented data with the references, as the number of cited publications approaches two hundred… I suggest a deep critical analysis of this part of text, especially please consider citing so many other review articles. I believe that the purpose of citing another publication in manuscripts should be a discussion with these. It is impossible for the reader to peek at over a hundred articles.

Round 2

Reviewer 2 Report

Thank You for Your answers, Ihave no further comments.